# S-PIFu: Integrating Parametric Human Models with PIFu for Single-view Clothed Human Reconstruction

**Kennard Yanting Chan**
S-Lab, Nanyang Technological University
`kenn0042@e.ntu.edu.sg`

**Guosheng Lin**
Nanyang Technological University
`gslin@ntu.edu.sg`

**Haiyu Zhao**
SenseTime Research
`zhaohaiyu@sensetime.com`

**Weisi Lin**
S-Lab, Nanyang Technological University
`wslin@ntu.edu.sg`

## Abstract

We present three novel strategies to incorporate a parametric body model into a pixel-aligned implicit model for single-view clothed human reconstruction. Firstly, we introduce ray-based sampling, a novel technique that transforms a parametric model into a set of highly informative, pixel-aligned 2D feature maps. Next, we propose a new type of feature based on blendweights. Blendweight-based labels serve as soft human parsing labels and help to improve the structural fidelity of reconstructed meshes. Finally, we show how we can extract and capitalize on body part orientation information from a parametric model to further improve reconstruction quality. Together, these three techniques form our S-PIFu framework, which significantly outperforms state-of-the-arts methods in all metrics. Our code is available at https://github.com/kcyt/SPIFu.

## 1   Introduction

Human digitization is an important topic that has application in areas like virtual reality, robot navigation, and medical imaging. While there exist high-end, multi-view capturing systems that can accurately digitalize and reconstruct 3D human bodies (4; 5; 13), they are largely inaccessible to general consumers. In search of more accessible methods, the research community has delved into human digitization approaches that use simple inputs such as a single image (2; 16; 22; 21).

Various methods have been proposed for single-image clothed human digitization, but one class of approaches that has gained notable attention is pixel-aligned implicit models (20; 21; 3), which are relatively lightweight and could capture fine-level geometric details such as clothes wrinkles. But these models are prone to depth ambiguity (causing unnaturally elongated body parts) and to producing human meshes with broken limbs (8; 18).

In order to overcome these two limitations, He *et al.* (7) designed ARCH++, a model which combines a pixel-aligned implicit model with a parametric human body model. However, ARCH++ depends heavily on having a near-perfect mapping between the canonical and posed spaces (especially during training), which is often infeasible to attain due to the varied clothing of human meshes. Another problem with ARCH++ is that it only uses the (x,y,z) coordinate information of the parametric model, ignoring other information that could well prove useful.

In order to overcome these issues, we propose S-PIFu, a pixel-aligned implicit model that utilizes a parametric human body model without requiring a mapping between the canonical and posed spaces. S-PIFu transforms a parametric body model into a set of pixel-aligned 2D feature maps, which can contain more information than just the (x,y,z) coordinate information. In its essence, S-PIFu is a new

36th Conference on Neural Information Processing Systems (NeurIPS 2022).

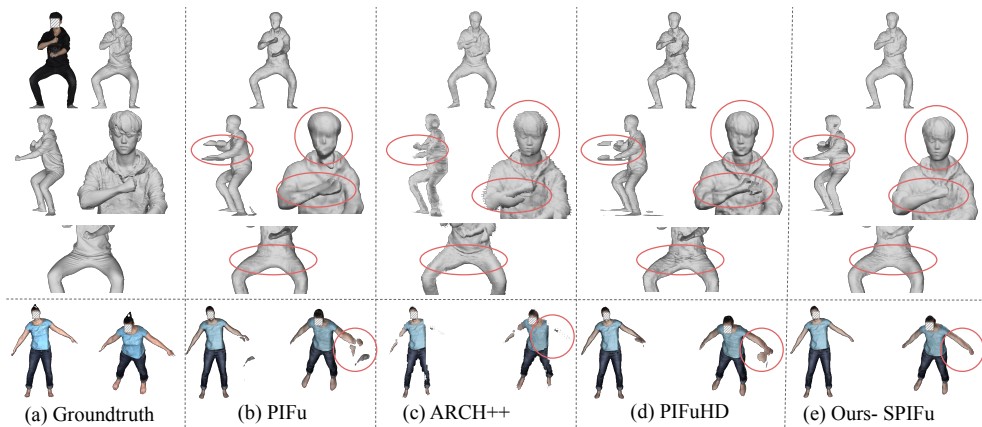

|     |     |     |     |     |
| --- | --- | --- | --- | --- |
| (a) Groundtruth | (b) PIFu | (c) ARCH++ | (d) PIFuHD | (e) Ours- SPIFu |

Figure 1: Compared with SOTA methods, including (b) PIFu (20), (c) ARCH++ (7) and (d) PIFuHD (21), our proposed model can precisely reconstruct the fine details of a human subject without floating artefacts or unnaturally elongated body parts. In the last row, the reconstruction outputs are colored by projecting the RGB values from the input image onto them. The purpose of the coloring is only to aid in visualization. Subject's face censored as required by dataset's owner.

method of incorporating a SMPL-X (17) parametric body model into a PIFu (which is a pixel-aligned implicit model). The 'S' stands for SMPL-X.

The main contributions of our work are:

1. We introduce ray-based sampling (RBS), a novel technique that transforms a parametric model into a set of highly informative, pixel-aligned 2D feature maps. We find that applying RBS on coordinate information alone can already propel a vanilla PIFu to SOTA performances.

2. Furthermore, we propose a new type of feature based on blendweights. Blendweight-based labels serve as soft human parsing labels and lead to a significant improvement over the baseline model.

3. Thirdly, we show how we can extract and capitalize on body part orientation information from a parametric model. We show this can also improve the baseline model significantly.

Each of our contributions can be added to any existing pixel-aligned implicit model.

## 2 Related Work

### 2.1 Single-view Human Reconstruction

Single-view human reconstruction (or digitalization) methods can be classified into parametric and non-parametric approaches. Parametric methods, such as (10; 12; 11), recover human body shapes by estimating the parameters of a parametric human body model (e.g. SMPL (14) or SMPLX (17)) from an input image. However, these parametric methods can only reconstruct clothless human meshes.

On the other hand, non-parametric approaches do not require a parametric body model. For example, (22) utilizes a 3D CNN to directly estimate a volumetric body shape from a single RGB image. A subclass of non-parametric approaches that have generated significant interest is pixel-aligned implicit models (20; 8; 3; 21). These methods model a human body as an implicit function, from which a mesh can be obtained using Marching Cubes algorithm (15). One of the first pixel-aligned implicit models is the Pixel-Aligned Implicit Function (PIFu) (20). PIFu uses a multi-layer perceptron to estimate the occupancy of query points in a 3D space. Using Marching Cubes, PIFu can reconstruct clothed human meshes effectively. A short illustration of PIFu is shown in Fig. 2 (See dotted box).

However, as aforementioned, pixel-aligned implicit models are prone to generating human meshes with broken limbs or unnaturally elongated human body parts. Recent pixel-aligned implicit models like IntegratedPIFu (3) propose techniques that successfully mitigate these problems, but the problems still exist. In contrast, parametric methods do not suffer from these problems. This observation led to hybrid methods that combine a pixel-aligned implicit model with a parametric method.

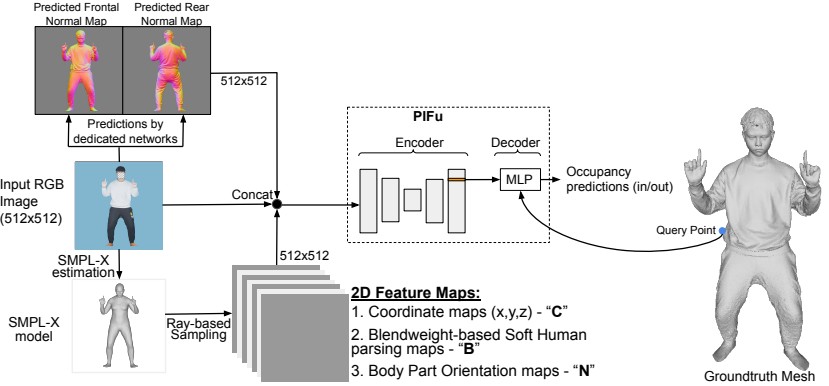

Figure 2: Illustration of our S-PIFu framework.

## 2.2 ARCH++

ARCH (9) is a hybrid method that uses a parametric body model to determine how a posed human mesh (posed space) could be mapped into a common pose (canonical space). A PIFu is then used to train and infer a body mesh in the canonical space. Another hybrid method, PaMIR (26), voxelizes an estimated SMPL (14) parametric body model before passing it through a 3D encoder to get voxel-aligned features, which are passed into a PIFu.

ARCH++ (7) improves upon ARCH and PaMIR by jointly estimating the occupancy of points in both the posed and canonical spaces. In addition, ARCH++ proposes using a PointNet++(19) to capture and encode the structural information of a parametric model. But, as aforementioned, ARCH++ relies heavily on having near-perfect mapping between the canonical and posed spaces and only uses the (x,y,z) coordinate information of the parametric model, ignoring other meaningful information.

## 2.3 Novelty of our Method

Our paper introduces Ray-based sampling (RBS), which is a completely different way of extracting information from a SMPL-like mesh. RBS directly generates pixel-aligned features from a SMPL-like mesh using rays that penetrate through the mesh. While the ray penetrates through the mesh, it collects meaningful information from the faces that it hit. We will elaborate on this later.

Compared with ARCH and ARCH++, RBS does not need the hard-to-obtain near-perfect mapping between canonical space and posed space (infeasible when human subject wears loose clothing like jacket or coat). Compared with PaMIR, RBS can generate features at much higher resolution (For height and width, RBS features has a 512x512 resolution, which is 16 times bigger than the 128x128 voxel features used in PaMIR). The resolution of PaMIR's voxel features is limited to 128x128 due to GPU memory constraints (26).

Moreover, we also introduce blendweight-based labels and body orientation labels (See later).

## 3 Method

An illustration of our S-PIFu is shown in Fig. 2. Similar to (21) and (7), we use a pix2pixHD network (23) to predict frontal and rear normal maps from an input RGB image. In addition, S-PIFu uses SMPLify-X (17) to estimate a SMPL-X parametric body model (17) from the input RGB image.

In contrast to PaMIR and ARCH++, our method does not need to train a separate encoder to encode a parametric model. Instead, we directly transform a parametric model into a set of 2D feature maps (called coordinate maps) using what we call ray-based sampling (See Section 3.1). The set of 2D feature maps is significantly more lightweight than voxels used by PaMIR, and, unlike ARCH and ARCH++, it does not require an error-free mapping between canonical and posed spaces.

Ray-based sampling captures (x,y,z) coordinate information from the SMPL-X model. But it can also be used to capture blendweight-based labels that serves as a proxy to soft human parsing labels (See

Section 3.2). The blendweight-based labels are represented as another set of 2D feature maps (called blendweight-based maps).

Beside soft human parsing labels, S-PIFu also leverages on ray-based sampling to extract body part orientation information from a SMPL-X model (See Section 3.3), giving us an additional set of 2D feature maps called body part orientation maps.

We concatenate coordinate maps, blendweight-based maps, body part orientation maps, frontal and rear normal maps, and a RGB input image together before feeding them into a PIFu, which will output occupancy predictions of query points in a 3D space. The PIFu will be used to reconstruct clothed human meshes, which are then refined with our normal maps using the mesh refinement technique employed in (7).

## 3.1 Ray-based Sampling

After estimating a SMPL-X model from a RGB image, we use Ray-based Sampling (RBS) to transform the SMPL-X model into a set of 2D feature maps. See an illustration of RBS in Fig. 3. In RBS, we project the SMPL-X model into an image plane. We interpret the projection lines as rays (in blue in Fig. 3) that move towards the SMPL-X, and each ray is associated with a pixel location.

Any ray that hits the SMPL-X mesh will simply penetrate through the mesh and continue moving in its original direction. Each ray will record the faces (more specifically, the face indices) of the mesh that it has penetrated through. Using the face indices, we can identify the SMPL-X vertices that form the penetrated faces. The (x,y,z) values of the vertices are then placed in 2D feature maps.

To be clearer, we will look at an example. Let us denote the **lower** ray (in blue) illustrated in 'Close-up' of Fig. 3 as $r$. We see that ray $r$ penetrates 4 faces ($f_1$, $f_2$, $f_3$, $f_4$) of the SMPL-X mesh. Each face in SMPL-X is formed by three vertices. Thus, we can identify the 12 vertices that form $f_1$, $f_2$, $f_3$, and $f_4$. If $f_1$ is formed by vertices $v_{11}$, $v_{12}$, and $v_{13}$, then we will average up the coordinates of $v_{11}$, $v_{12}$, and $v_{13}$ to obtain the face average coordinates $\bar{x}_1$, $\bar{y}_1$, $\bar{z}_1$. With 4 faces, the ray $r$ will have a total of 4 sets of face average coordinates. Recall that each ray is associated with exactly one pixel location in the image plane (the image plane actually corresponds to a 2D feature map). Thus, we will need 4x3 or 12 2D feature maps to store the information captured by the ray (4 refers to the number of face average coordinates, and 3 refers to the x,y,z coordinates). *Terminology: In some literature, the 12 feature maps are also referred to as a feature map with 12 channels.*

We limit the number of faces that a ray can penetrate through to 6 faces as it is very unlikely for a ray to penetrate more than 6 faces. In total, RBS will compute 6x3 or 18 2D feature maps (if we only use coordinate information). If we specify the 2D feature maps to have a resolution of 512x512, then a total of 512x512 rays will be used. This is actually relatively lightweight in practice because RBS only need to be done once, and it can be done in pre-training. The 2D feature maps can be efficiently stored using sparse matrices as a significant number of rays will not hit the SMPL-X mesh at all (Also, this means that zeroes will be inserted onto a pixel on the 2D feature map if a ray that corresponds to that pixel does not hit any surface.)

PaMIR uses memory-intensive 3D voxels that are prone to quantization errors caused by the necessary use of low spatial resolution (to preserve memory) (7). Compare to voxels, our set of 2D feature maps is significantly more lightweight, making it possible to utilize a much higher resolution of the maps.

In addition, unlike ARCH++, RBS does not require a mapping between canonical and posed spaces and is thus not prone to errors in the mapping. In Section 4.2.2, we show that RBS with coordinate information alone is able to significantly outperform ARCH++ in almost all metrics.

In short, RBS generates 18 2D feature maps from a SMPL-X model. These feature maps are concatenated with the original inputs of a PIFu and then fed into the PIFu as originally designed. The 18 feature maps inform the PIFu of the structural information of a SMPL-X mesh.

## 3.2 Using Blendweight-based labels as a proxy to Soft Human Parsing Labels

In existing hybrid models, such as ARCH (9), PaMIR (26), and ARCH++ (7), the only information that was extracted from the predicted parametric human body model is the (x,y,z) coordinate information. It is a waste as a parametric body model contains much more information than just the (x,y,z)

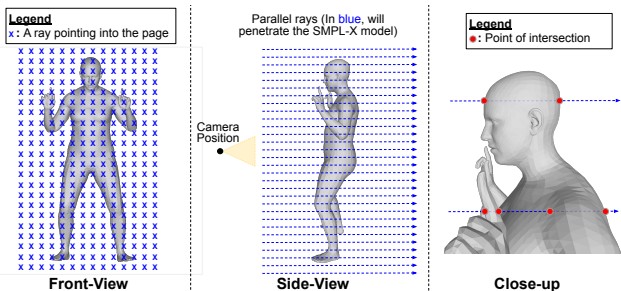

Figure 3: Illustration of Ray-based Sampling. As in (20; 9; 7), a weak-perspective camera is assumed. *The diagram assumes the SMPL-X model has been scaled, thus rays and projection lines are parallel.*

coordinate information. One such information is the human parsing information contained in a parametric body model.

Parametric human models, such as SMPL(14) and SMPL-X(17), are formed by a predefined and fixed number of vertices. Each of these vertices can be easily mapped to a discrete human parsing label (e.g. head, upper left arm, hip etc.). While the pose and shape of a parametric model can change, the discrete human parsing label for each of these vertices remain largely unchanged. For example, if the 50th vertex of the SMPL-X model has a human parsing label of 'head', it will retain this human parsing label regardless of what pose and shape the SMPL-X model takes.

Using Ray-based Sampling (RBS), we can map the SMPL-X vertices to pixel locations of a 2D feature map. Since we know the discrete human parsing label of each SMPL-X vertex, we can essentially create a 2D feature map that consists of the discrete human parsing labels that are given to each SMPL-X vertex. Please refer to Fig. 4a on more detail on how the 2D feature map is computed. This 2D feature map can then be fed into a PIFu. (In reality, as explained in Section 3.1, we need a total of 6 such 2D feature map because a ray can penetrate up to 6 faces.)

However, assigning a discrete human parsing label to each SMPL-X vertex can cause unnecessary noise at the boundary between two different human body parts. For example, looking at Fig. 4a, we observe that at the boundary between two different body parts, the vertices can be assigned two very different human parsing labels despite being very similar to one another. Moreover, vertices that are assigned the same human parsing label (e.g. lower left leg), can actually be very different from one another.

In order to resolve these shortcomings, we would require a method to give a form of soft human parsing labels to each of the SMPL-X vertices. This leads us to the concept of blendweight in the SMPL-X model. Blendweight is a matrix of shape (N x K) where N is the total number of SMPL-X vertices, and K is the number of joints in SMPL-X. Essentially, the blendweight matrix contains a K-length vector of values for each SMPL-X vertex. This vector determines how much each joint in SMPL-X would influence the position of a specific vertex when the SMPL-X model is transformed from a default pose to an arbitrary pose. We can interpret this K-length vector as a vector of weights that determine the degree of association that a specific SMPL-X vertex has with the K different joints in the SMPL-X body. In other words, if we consider joint locations to be a rough proxy for body part locations, then we can use this K-length vector as a soft human parsing label for a vertex.

A simplistic visualization of how blendweight can be used as soft human parsing labels is shown in Fig. 4b. In reality, the blendweight is more complicated (and useful) than that as each vertex is given a K-length vector where K=55 in SMPL-X model. Thus, unlike what is shown in the figure, the soft human parsing label given to a vertex does not merely weigh the association of that vertex with two different body parts (or joints). The soft human parsing label weighs the association of that vertex with 55 different joints. In a broader sense, blendweight-based labelling is simply giving each vertex an unique ID, except that the ID is made up of mathematically meaningful values that tell us how a vertex is similar to another vertex and in what aspects are they similar (e.g. Are two vertices both influenced by the 10[th] joint?). In addition, we provide a comparison between using discrete human parsing labels and blendweight-based soft parsing labels in supplementary materials.

In order to reduce computational cost, we utilize principal component analysis (PCA) to reduce the dimensions of the blendweight matrix from (N x 55) to (N x 24). Thus, we can map each vertex to a 24-length vector. Using RBS, we will generate 6 x 24 or 144 2D feature maps (As explained in

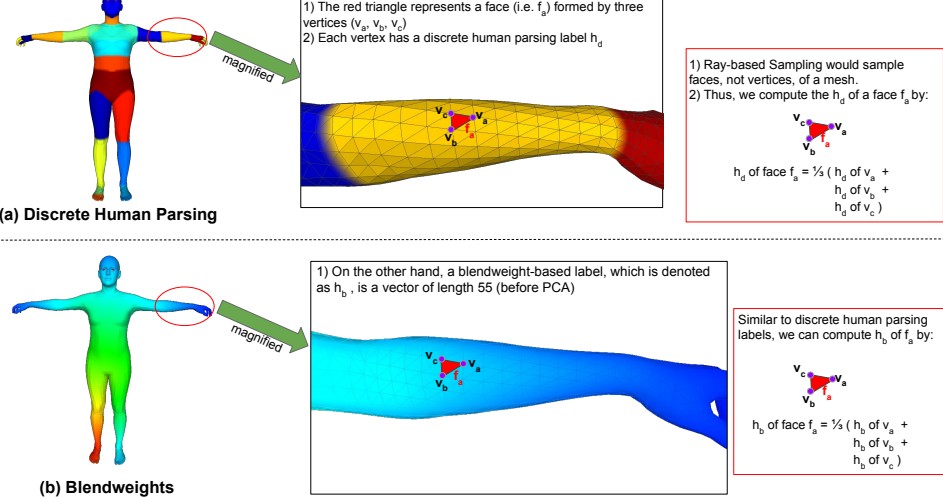

Figure 4: Comparison of Discrete Human Parsing labels with Blendweight Human Parsing labels. In (a), the $h_d$ of a vertex is a one-hot vector that has length of 27 if we decide to partition the human body into 27 different body parts. In (b), each vertex has a $h_b$ label. Each element in a $h_b$ vector is a real number (unlike a discrete human parsing label, which is a one-hot vector)

Section 3.1, a ray can penetrate up to 6 faces.). An illustration on how we compute the 2D feature maps is shown in Fig. 4b.

### 3.3 Encoding the Orientation of Body Parts using Vertex Normal information

With blendweight-based labels, our model will have fine-grained body part labels of a human body. However, in both 2D and 3D spaces, the same body part can appear very different when it is oriented differently. For example, a closed fist and a wide open hand appear very differently despite belonging to the same body part. This limits the usefulness of having human parsing labels (even if it is soft or fine-grained labels) as any model would be confused by how the exact same human parsing label can have vastly different appearances. Thus, we argue that it is important for human parsing labels to be complemented with body part *orientation* information.

To extract body part orientation information from a predicted SMPL-X model, we start by applying Ray-based Sampling (RBS). For each SMPL-X face $f$ that is penetrated by a ray, we collect the vertex normal vectors of the three vertices that constitute that face. We then compute the face average normal of $f$ by taking the average of the three vertex normal vectors. This face average normal vector tells us the current orientation of the body part (more accurately, the orientation of a face on that body part). In order to provide our model with more information about the body parts, we also provide the initial orientation of the body parts (i.e. the face average normal of the same face $f$ but when the SMPL-X model is in the initial or rest pose. See Fig. 5 for an illustration). The reason for including the initial orientation is to allow our model to compute how different the current orientation (of a body part) is from its initial orientation. If the difference is small, it tells our model that the body part should appear similar to how it appears at rest pose. But if the difference is big, the body parts should appear more different from its rest pose. (Refer to the short example in Fig. 5) In other words, to help our model interpret face average normals, we are giving our model a reference face average normal for every face in the SMPL-X.

In addition, the inclusion of face normals in RBS is important because, in RBS, a ray does not know whether a face it penetrated is a face that is facing towards the camera or a face that is facing away from the camera. In the context of a PIFu, which is the base architecture of our S-PIFu, it is important to know whether a face is facing towards or away from the camera. This is because PIFu works by predicting whether a query point is inside or outside a mesh. So when there is a face that is facing towards the camera, it means that query points in front of this face is outside of the mesh. Likewise, when there is a face facing away from camera, it means query points in front of the face surface are inside of the mesh. The face normals thus provide important hints to PIFu to aid its predictions.

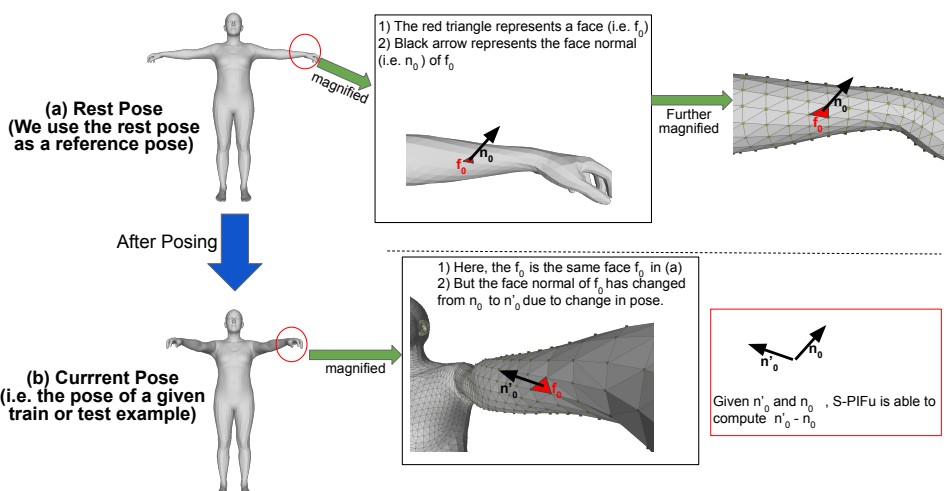

Figure 5: Illustration of the change in orientation for a face on a SMPL-X body model. $n'_0 - n_0$ tells S-PIFu how far a face's orientation has been changed when a rest-pose mesh (a) is being posed to the current pose (b). Thus, $n'_0 - n_0$ does contain information about the current pose (i.e. if $n'_0 - n_0$ is equal to a zero vector, then current pose is the rest pose.)

# 4    Experiments

## 4.1    Datasets

We train our models and the competing models on the THuman2.0 dataset (24), which consists of high-quality scans of human subjects. The subjects are primarily young Chinese adults. We exclude scans or meshes that have very obvious scanning errors. Meshes that suffer from extreme self-occlusion are also excluded because we do not hope to reduce the task into a guessing game. This data cleaning process is conducted before any of the models is trained so as to ensure fair comparison between the different models. In all, we use 362 human meshes from the THuman2.0 dataset. A 80-20 train-test split is used, and for each mesh, we render 10 RGB images at different yaw angles with a weak-perspective camera.

In addition, we include the BUFF dataset (25) as an additional evaluation dataset. No model is trained using the BUFF dataset. Systematic sampling based on sequence number is carried out on the BUFF dataset to obtain a total of 93 human meshes for testing. We use systematic sampling to ensure that, for the same human subject, meshes of different poses are obtained and repeated (or near identical) poses are omitted. More implementation details is included in our supplementary materials.

## 4.2    Comparison with State-of-the-art

We compare S-PIFu against the existing models on single-view clothed human reconstruction qualitatively and quantitatively. For our quantitative evaluation, we follow the same metrics as used in (20; 21; 7). Specifically, we use Chamfer distance (CD), Point-to-Surface (P2S), and Normal reprojection error (Normal).

Existing methods that we compared against include PIFu (20), ARCH++ (7), and PIFuHD (21). We included PIFu as it is our base architecture and thus serve as an important baseline. ARCH++ is used as it is the state-of-the-arts for incorporating a parametric human body model into a pixel-aligned implicit model. Also, in (7), He *et al.* demonstrated that ARCH++ is an improvement over both ARCH (9) and PaMIR (26), and this is our explanation for not including ARCH and PaMIR. In addition, another reason for not including PaMIR is that its authors did not demonstrate that PaMIR is able to outperform PIFuHD, which is the state-of-the-arts at that time. We include PIFuHD as it is still one of the most effective methods for single-view clothed human reconstruction.

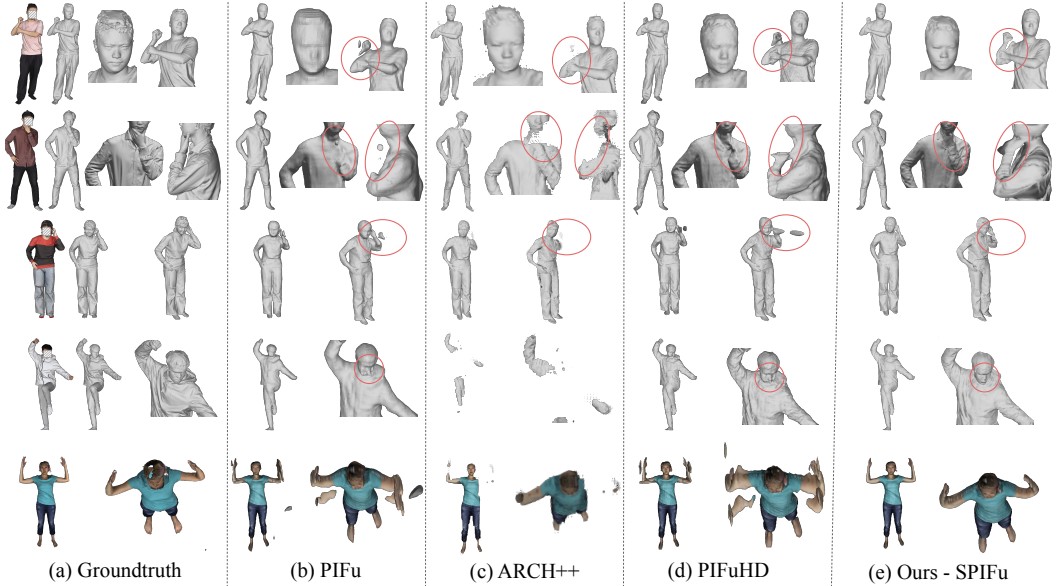

| (a) Groundtruth | (b) PIFu | (c) ARCH++ | (d) PIFuHD | (e) Ours - SPIFu |

Figure 6: Qualitative evaluation with state-of-the-art methods, including (b) PIFu (20), (c) ARCH++ (7) and (d) PIFuHD (21). For each row, the first object is the input RGB image given to the models.

### 4.2.1 Qualitative Evaluation

We present a qualitative comparison of our S-PIFu with the other existing methods in Fig.1 and Fig.6. In Fig.1, we observe that only our approach is able to capture the correct structure of the human subject body. Other methods tend to produce unnaturally elongated body parts, missing body parts, or floating artefacts. Moreover, S-PIFu successfully reconstructs the fine details of the human body, such as the wrinkles on the long pants and the precise facial features.

In Fig.6, we observe that PIFu, ARCH++, and PIFuHD tend to generate broken limbs, floating artefacts, or missing body parts. In contrast, S-PIFu is able to generate a clean, natural structure of human body. We believe that ARCH++'s tendency to produce missing body parts is due to the imperfect mapping between the canonical and posed spaces. In (7), ARCH++ is trained using the RenderPeople dataset (1), which mainly consists of meshes in upright standing poses (6). THuman2.0 dataset, however, has more varied and complicated poses, making perfect mapping harder to achieve.

### 4.2.2 Quantitative Evaluation

In addition, we provide a quantitative evaluation of our methods against the state-of-the-arts methods in Tab. 1. The table shows that when a PIFu is incorporated with coordinates information extracted using our ray-based sampling ('PIFu + M + C'), it can achieve significant improvements and even outperforms current state-of-the-arts models in the THuman2.0 dataset. However, PIFu + M + C does not do as well in the BUFF dataset, indicating a lesser generalizability to a foreign dataset.

In contrast, S-PIFu, which combine coordinates information (C), blendweight-based labels (B), and body part orientation information (N) into a PIFu (together with frontal and rear normal maps M), outperforms the existing state-of-the-arts models in all metrics in both datasets. This suggests that the inclusion of B and N helps our model to learn more generalizable features from the training data.

### 4.3 Ablation Studies

### 4.3.1 Evaluation of using Ray-based Sampling to extract coordinate information

In order to evaluate the benefits of using Ray-based sampling (RBS) to extract coordinate information from a SMPL-X mesh, we train a PIFu that uses not only the RGB image, frontal normal map, and rear normal map, but also the additional input of coordinates (x,y,z) extracted via RBS. We compare this model with a vanilla PIFu that is trained with only the RGB image and the frontal and rear normal maps. See qualitative results in Fig. 7. The PIFu with coordinates information can better capture

Table 1: Quantitative evaluation of our models against the state-of-the-arts in the THuman2.0 test set and BUFF dataset. (M = frontal and rear normal maps. C,B,N = Coordinates information, Blendweight-based labels, Body part orientation information from Ray-based Sampling respectively. S-PIFu combines C, B, and N into a PIFu with M i.e. S-PIFu = PIFu + M + C + B + N )

| | THuman2.0 | | | BUFF | | |
|---|---|---|---|---|---|---|
| Methods | CD ($10^{-4}$) | P2S ($10^{-4}$) | Normal | CD ($10^3$) | P2S ($10^3$) | Normal |
| PIFu | 10.83 | 6.671 | 2.211 | 3.427 | 3.926 | 2.580 |
| PIFu + M | 3.085 | 2.673 | 1.731 | 1.979 | 1.973 | 1.947 |
| ARCH++ | 3.670 | 1.979 | 2.131 | 3.050 | 1.811 | 3.436 |
| PIFuHD | 2.758 | 2.215 | 1.698 | 1.964 | 1.845 | 2.010 |
| (Ours) PIFu + M + C | **1.999** | **1.626** | **1.413** | 2.022 | 1.840 | **1.536** |
| (Ours) PIFu + M + B | 2.595 | 2.079 | 1.411 | 2.050 | 1.866 | 1.559 |
| (Ours) PIFu + M + N | 2.818 | 2.275 | 1.416 | 2.060 | 1.931 | 1.557 |
| (Ours) S-PIFu | 2.035 | 1.629 | 1.440 | **1.923** | **1.726** | 1.564 |

the structure of human meshes compared to the other PIFu, which often produces broken or missing limbs, and floating artefacts.

Quantitatively, we can compare the CD and P2S of the two models using the second row ('PIFu + M') and the fifth row ('PIFu + M + C') of Tab. 1. We find that although 'PIFu + M + C' fared marginally worse than 'PIFu + M' in the CD of BUFF dataset, it significantly outperformed 'PIFu + M' in the CD and P2S of the THuman2.0 dataset and in the P2S of the BUFF dataset.

As mentioned in Section 4.2.2, 'PIFu + M + C' also outperforms ARCH++ and other methods.

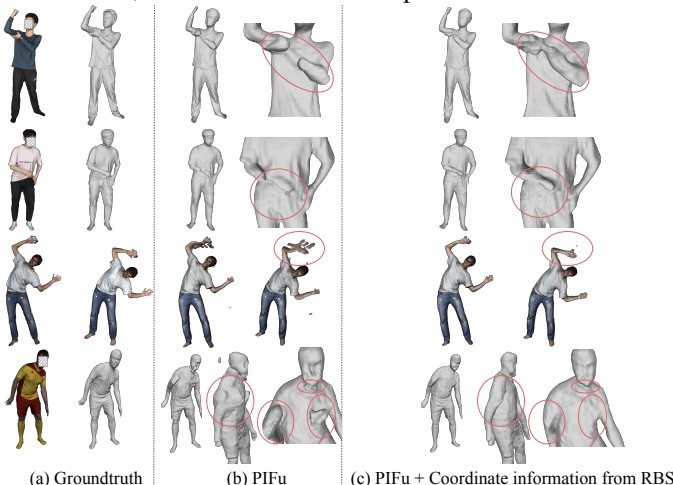

(a) Groundtruth        (b) PIFu        (c) PIFu + Coordinate information from RBS

Figure 7: Effect of incorporating SMPL-X coordinates extracted using Ray-based sampling (RBS).

### 4.3.2 Evaluation of using Blendweight-based labels

Similarly, in order to evaluate the effect of using Blendweight-based labels, we train a PIFu that takes in Blendweight-based labels as the only additional inputs (besides RGB image and frontal and rear normal maps). We compare this model with a PIFu that is given only the RGB image and the frontal and rear normal maps. See the qualitative results in Fig. 8. We find that using blendweight-based labels helps to improve the structural fidelity of reconstructed meshes. Specifically, it tackles the problems of unnaturally elongated human parts and floating artefacts.

Quantitatively, we can compare the CD and P2S of the two models using the second row ('PIFu + M') and the sixth row ('PIFu + M + B') of Tab. 1. Similarly, we find that 'PIFu + M + B' only fared marginally worse than 'PIFu + M' in the CD of BUFF dataset. For the CD and P2S in THuman2.0 and P2S in BUFF, 'PIFu + M + B' significantly outperformed 'PIFu + M'. Also, in our Supp. Mat., we show the results when we replace the 'B' in 'PIFu + M + B' with discrete human parsing labels.

### 4.3.3 Evaluation of using Body Part Orientation Information

Lastly, we evaluate the merits of using body part orientation information by comparing a PIFu that is given only RGB image and frontal and rear normal maps with another PIFu that is given the additional

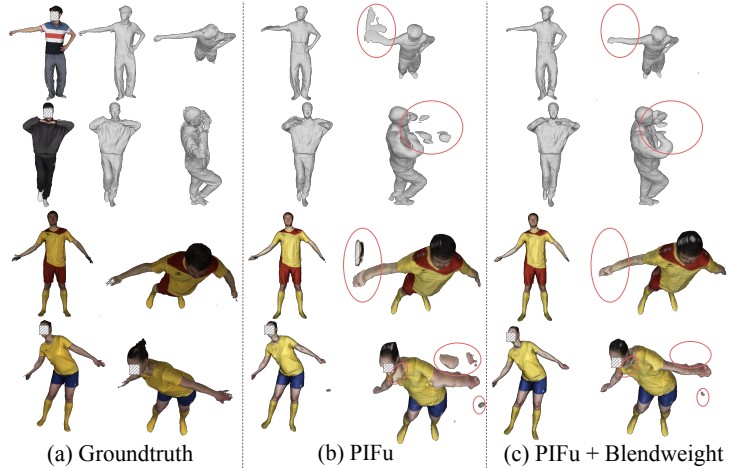

|  (a) Groundtruth | (b) PIFu | (c) PIFu + Blendweight |

Figure 8: Evaluation of the effect of incorporating Blendweight-based Human Parsing Information

input of body part orientation information. See the qualitative results in Fig. 9. We find that using body part orientation information helps to further reinforce the structural accuracy of reconstructed meshes by mitigating the occurrence of noisy artefacts in the reconstruction outputs.

Quantitatively, we can compare the CD and P2S of the two models using the second row ('PIFu + M') and the seventh row ('PIFu + M + N') of Tab. 1. Except for the CD in Buff dataset, for which 'PIFu + M + N' performed marginally worse than 'PIFu + M', 'PIFu + M + N' significantly outperformed 'PIFu + M' in the CD and P2S of the two datasets. Also, in our Supp. Mat., we compare the results when we replace the 'N' in 'PIFu + M + N' with 'N' but without the initial orientation of body parts.

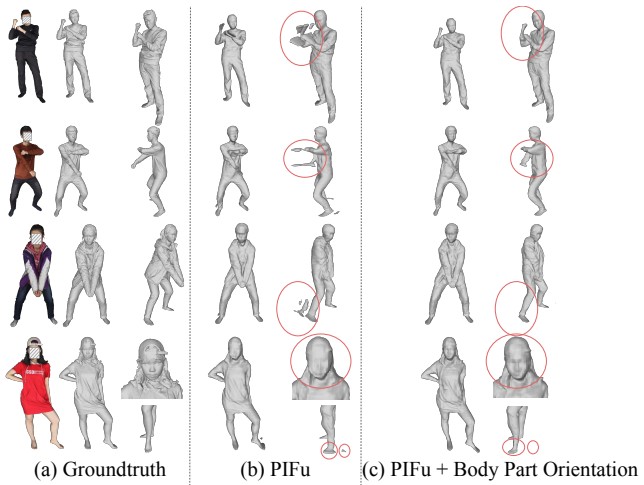

|  (a) Groundtruth | (b) PIFu | (c) PIFu + Body Part Orientation |

Figure 9: Evaluation of the effect of incorporating Body Part Orientation Information

## 5   Conclusion

Our S-PIFu framework consists of three novel strategies designed to incorporate a parametric human body model into a pixel-aligned implicit model. Firstly, we proposed ray-based sampling, a novel technique that transforms a parametric model into highly informative 2D feature maps. Next, we introduced a new type of feature based on blendweights. Finally, we showed how body part orientation information can be extracted from a SMPL-X model to further improve reconstruction quality.

## Acknowledgments and Disclosure of Funding

This study is supported under the RIE2020 Industry Alignment Fund – Industry Collaboration Projects (IAF-ICP) Funding Initiative, as well as cash and in-kind contribution from the industry partner(s). G. Lin's participation is supported by the Ministry of Education, Singapore, under its Academic Research Fund Tier 2 (MOE-T2EP20220-0007).

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
