# Supplementary Materials for S-PIFu: Integrating Parametric Human Models with PIFu for Single-view Clothed Human Reconstruction

**Kennard Yanting Chan**
S-Lab, Nanyang Technological University
kenn0042@e.ntu.edu.sg

**Guosheng Lin**
Nanyang Technological University
gslin@ntu.edu.sg

**Haiyu Zhao**
SenseTime Research
zhaohaiyu@sensetime.com

**Weisi Lin**
S-Lab, Nanyang Technological University
wslin@ntu.edu.sg

## 1 Performance of S-PIFu on pixels that do not belong to the estimated SMPL-X body.

In Fig. 1, we show S-PIFu's results when given images of test subjects who wear large clothings (e.g. jackets/coats). Images of these test subjects have pixels that belong to human subject but not to the SMPL-X body, and yet S-PIFu is able reconstruct the human subjects accurately.

Pixels that belong to human subject but not to the SMPL-X body act as a natural regularizer that prevents S-PIFu from being overly reliant on estimated SMPL-X meshes to reconstruct clothed human meshes. This happens because these pixels only have valid values for the RGB channels and not the channels of our 2D feature maps (i.e. C, B, and N. Recall that C refers to coordinate information, B refers to blendweights-based labels, and N refers to body part orientation information).

## 2 Robustness of S-PIFu to inaccurate underlying SMPL-X fitting.

In Fig. 2, we observe what would happen if we feed a noisy SMPL-X mesh (i.e. a SMPL-X mesh with inaccurate pose parameters) to our S-PIFu (Note that S-PIFu has not been trained with any noisy SMPL-X meshes). It is not uncommon for an estimated SMPL-X mesh to have an inaccurate pose, as observed by PaMIR (9), ARCH++ (3) and ICON (8).

In the first row, we obtained the noisy SMPL-X mesh by shift the position of the pseudo-groundtruth SMPL-X mesh's arms (both arms). From the frontal view, it appears that S-PIFu is unaffected by this. But on a closer examination, we notice that the mesh reconstructed by S-PIFu has a slight change in shape at its right arm. However, there is no noticeable change at its left arm. For the second row, we rotate the pseudo-groundtruth SMPL-X mesh about its y-axis. Again, we notice that S-PIFu is indeed affected by the noisy SMPL-X mesh, but the impact remains limited.

For the third row, we tucked the SMPL-X mesh's hands close onto its body and perform both rotation (about y-axis) and translation (in x-direction). Despite the drastic changes made to the SMPL-X mesh, S-PIFu is able to ignore incorrect information provided by the noisy SMPL-X. The only noticable change is slight change in the reconstructed mesh's left cheek.

For the fourth row, we move the SMPL-X mesh's four limbs and also its neck. Surprisingly, we cannot find any discernible abnormality in the mesh reconstructed by S-PIFu. This demonstrates the robustness of our method. S-PIFu seems to be aware of noises in the given SMPL-X's pose and is able to adjust accordingly.

36th Conference on Neural Information Processing Systems (NeurIPS 2022).

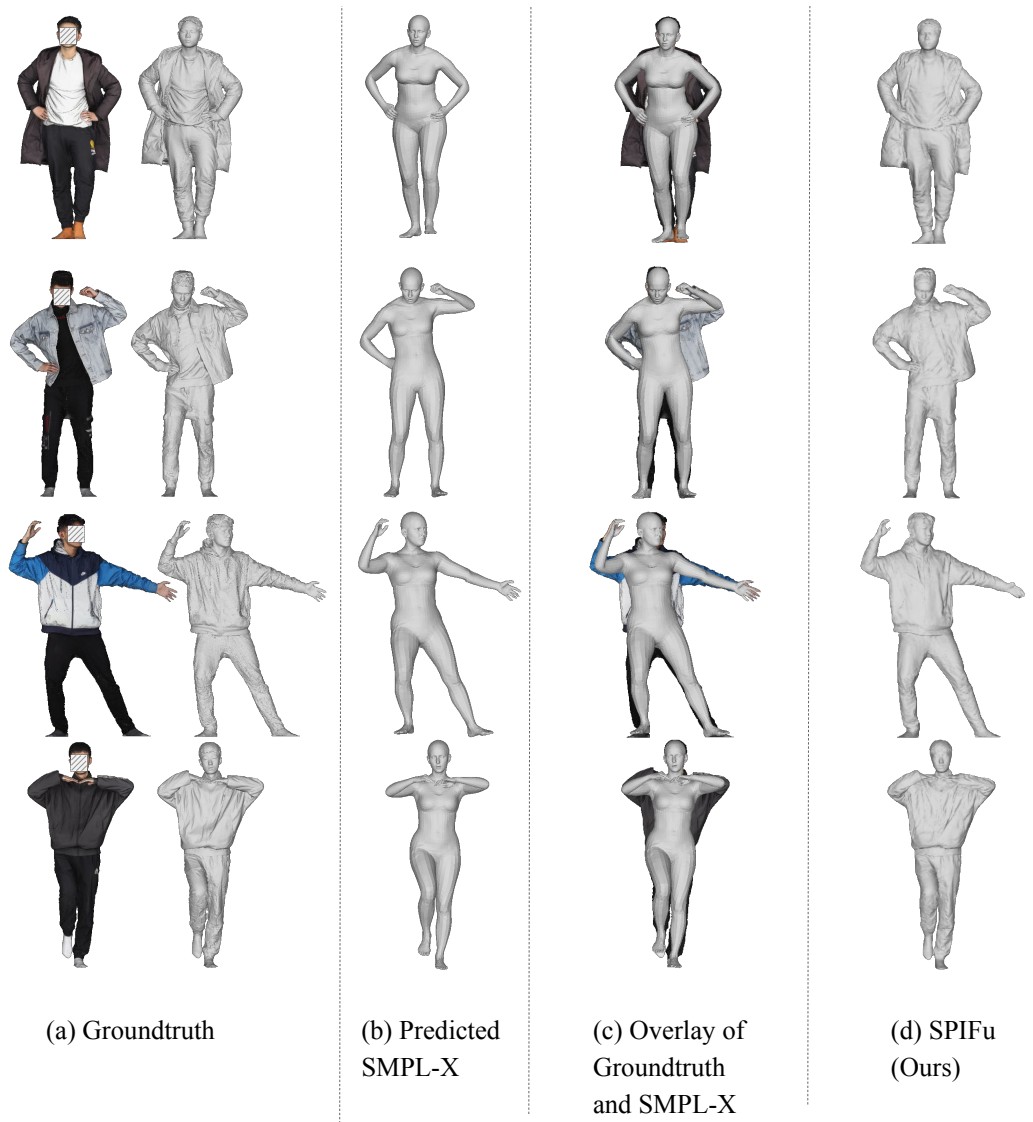

|  (a) Groundtruth | (b) Predicted SMPL-X | (c) Overlay of Groundtruth and SMPL-X | (d) SPIFu (Ours) |

Figure 1: Evaluation of S-PIFu on Human Subjects with thick clothing. (b) is predicted by SMPLify-X (and then finetuned using OpenPose keypoints, we will elaborate on this in Section3). (c) is the overlay of (a) and (b). (d) is the meshes reconstructed by S-PIFu.

One reason why S-PIFu is robust to noisy SMPL-X meshes is that S-PIFu only makes use of the estimated SMPL-X at the start of our pipeline, giving the opportunity for S-PIFu to use the RGB input image to correct any errors in the estimated SMPL-X before S-PIFu's MLP estimates a final 3D occupancy volume and reconstructs a 3D clothed human mesh. In other words, S-PIFu is more robust to errors in the estimated SMPL-X compared to models like ICON (8), which make use of SMPL-X at the start and end of its pipeline.

## 3   Additional refinement of estimated SMPL-X pose parameters

After the SMPL-X estimation by SMPLify-X, we further refine the SMPL-X mesh's body pose using the strategy described by ARCH++ (3). In short, we use the keypoints predicted by OpenPose to further finetune the SMPL-X mesh's body pose. The distance (mean square error) between the Openpose keypoints and the SMPL-X mesh's joints are used to optimize the SMPL-X's body pose parameters. We illustrate the finetuning process in Fig. 3.

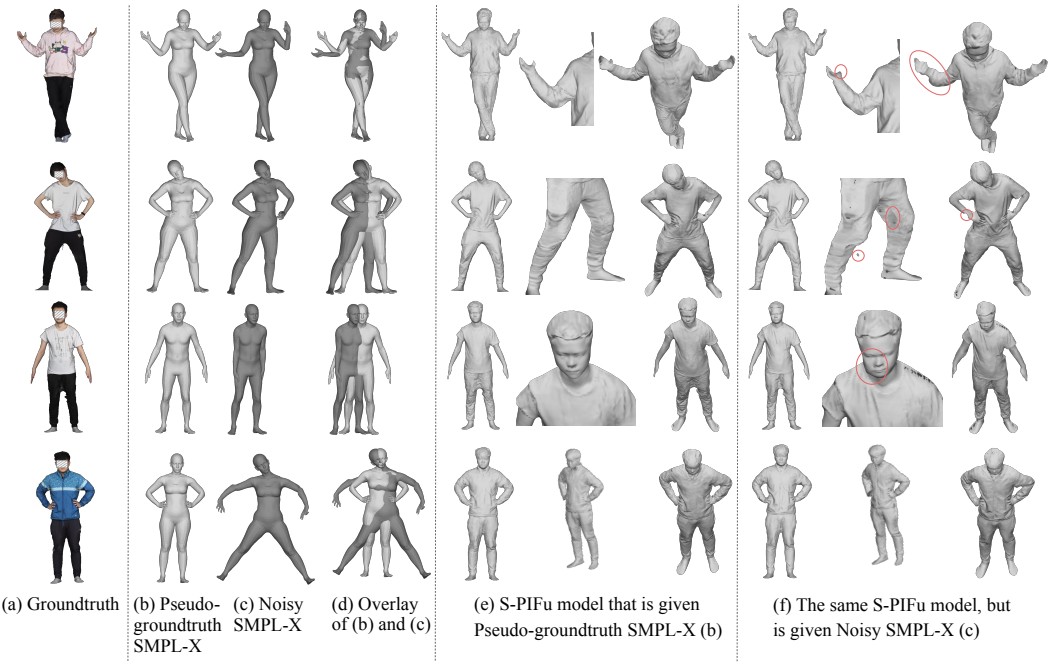

(a) Groundtruth    (b) Pseudo-groundtruth SMPL-X    (c) Noisy SMPL-X    (d) Overlay of (b) and (c)    (e) S-PIFu model that is given Pseudo-groundtruth SMPL-X (b)    (f) The same S-PIFu model, but is given Noisy SMPL-X (c)

Figure 2: Evaluation of the Robustness of S-PIFu to Inaccurate SMPL-X fittings. Red circles indicate the only noticeable changes that we could find. (b) Pseudo-groundtruth SMPL-X meshes are obtained by first predicting a SMPL-X mesh from an input RGB using SMPLify-X, and then finetuning the pose of that SMPLify-X mesh using OpenPose keypoints. More details of the finetuning process is given in Section 3.

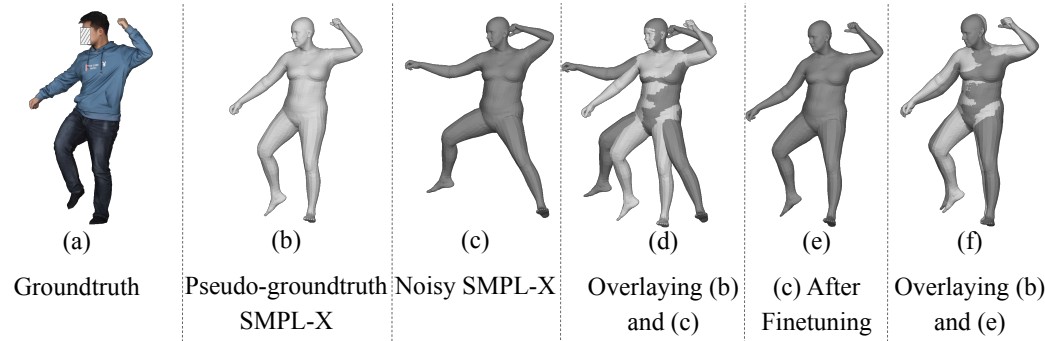

| (a) | (b) | (c) | (d) | (e) | (f) |
|-----|-----|-----|-----|-----|-----|
| Groundtruth | Pseudo-groundtruth SMPL-X | Noisy SMPL-X | Overlaying (b) and (c) | (c) After Finetuning | Overlaying (b) and (e) |

Figure 3: Illustration of how we finetune the body pose of a SMPL-X mesh estimated by SMPLify-X. The incorrectly estimated SMPL-X mesh (c) is transformed into (e) after finetuning.

## 4   Performance of S-PIFu on real images in the wild.

We compare PIFuHD (7) (as it was the best competing model in our paper) and S-PIFu with real Internet images sourced from Shutterstock in Fig. 4. We did not include more recent works like IntegratedPIFu (1) because that paper was published only after the NeurIPS 2022 deadline. IntegratedPIFu is an improvement over PIFuHD, but it is fundamentally still a pure pixel-aligned implicit model (i.e. does not use parametric human body model). Thus, while IntegratedPIFu is able to deal with problems such broken limbs and depth ambiguity better than PIFuHD, it still does not completely solve these problems.

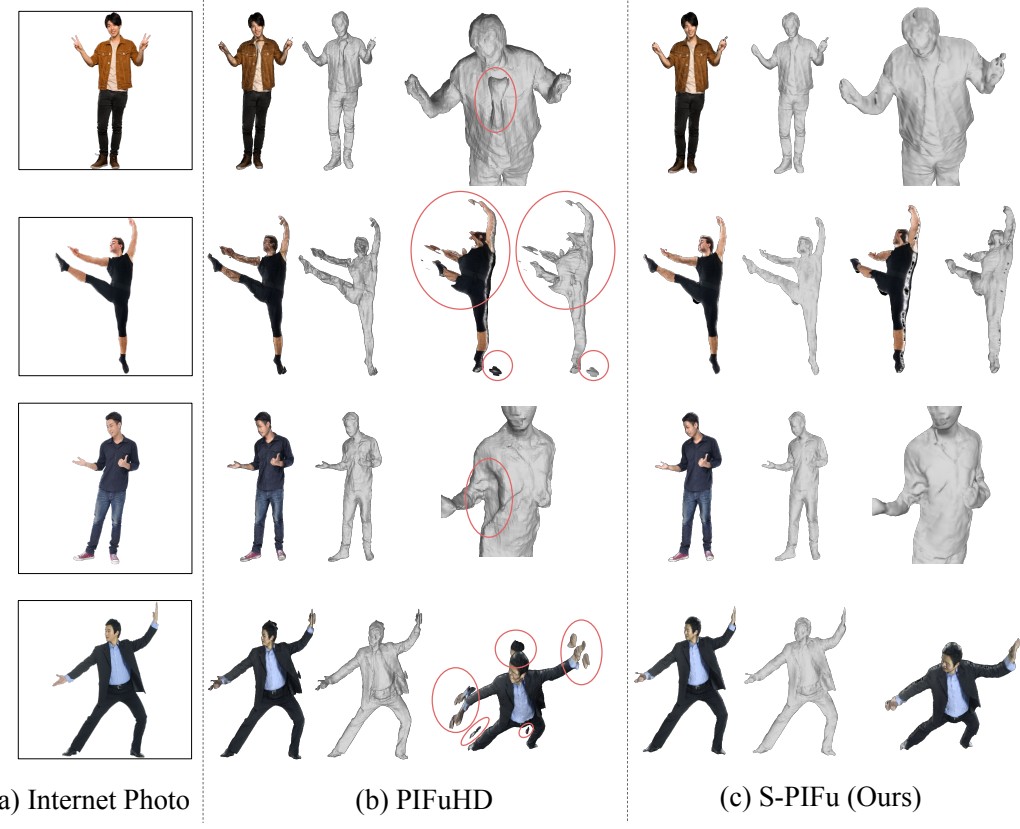

(a) Internet Photo    (b) PIFuHD    (c) S-PIFu (Ours)

Figure 4: Comparison of PIFu-HD and S-PIFu on real Internet photos sourced from Shutterstock.

# 5    Comparison of S-PIFu with ICON

Here, we evaluate and compare S-PIFu against ICON (8). As the ICON's paper and training script were released very close to the NeurIPS 2022 deadline, we are unable to include ICON in our main paper. But we do agree that ICON (CVPR 2022) is a paper that should be compared against our S-PIFu. Thus, we provided this section.

In this section, we trained and evaluated both ICON and S-PIFu on all 526 3D human scans from THuman2.0 dataset using a 80-20 train-test split. For the BUFF dataset, we use the same 93 scans that we had used in our main paper.

It is clear that ICON's and S-PIFu's architectures are very different, and there is no need for us to further highlight that. Another difference that we observed is that ICON tends to depend a lot more on the estimated SMPL-X compared to S-PIFu. This makes sense because ICON's final MLP, which is responsible for predicting the final 3D occupancy volume, uses the signed distance between a query point and the closest SMPL-X body point as an input. In contrast, S-PIFu only makes use of the estimated SMPL-X at the start of our pipeline, giving the opportunity for S-PIFu to use the RGB input image to correct any errors in the estimated SMPL-X before S-PIFu's own MLP estimates a final 3D occupancy volume. In other words, S-PIFu is more robust to errors in the estimated SMPL-X compared to ICON.

Quantitatively, we can refer to Tab. 1. We see that our S-PIFu outperformed ICON in all datasets, especially in the THuman2.0 test set. We believe the reason why ICON does poorly on the test set of THuman2.0 is that it over-relied on the estimated SMPL-X mesh. The estimated (and refined) SMPL-X mesh is consistently misaligned with the groundtruth mesh from the test set. For example, many groundtruth meshes in the THuman2.0 test set place their hands in their pockets. In such cases, we observe that the estimated (and refined) SMPL-X meshes rarely put their hands close to the pocket region, preferring to have their hands placed further away from their bodies (and closer to the camera). This misalignment in SMPL-X meshes caused ICON to under-perform in the THuman2.0

Table 1: Quantitative evaluation of S-PIFu against ICON (8). Both models are trained and evaluated on the entire 526 3D scans from THuman2.0 dataset with a 80-20 train-test split.

| | THuman2.0 | | | BUFF | | |
|---|---|---|---|---|---|---|
| Methods | CD ($10^{-4}$) | P2S ($10^{-4}$) | Normal | CD ($10^3$) | P2S ($10^3$) | Normal |
| ICON | 7.848 | 8.541 | 4.686 | 2.656 | 2.984 | 3.118 |
| (Ours) S-PIFu | **3.080** | **3.504** | **2.482** | **2.188** | **1.997** | **1.533** |

test set because the clothed human meshes reconstructed by ICON does not deviate too much from the estimated (and refined) SMPL-X meshes.

Qualitatively, we can refer to Fig. 5. As aforementioned, ICON suffers from over-reliance on the estimated (and refined) SMPL-X mesh. In many of the test examples, the SMPL-X mesh does not fully aligned with the groundtruth mesh, and this leads to ICON generating misaligned clothed human meshes. Another problem that ICON has is that ICON tends to produce noisy artefacts surrounding the limbs of the reconstructed clothed human meshes. Our S-PIFu does not have such problems.

In addition, we also provide an evaluation of ICON and our SPIFu using real Internet images sourced from Shutterstock. We provide the results in Fig. 6. ICON suffers from the same problems of misaligned SMPL-X meshes (leading to misaligned clothed human meshes) and noisy artefacts surrounding the limbs of clothed human meshes generated by ICON. Additionally, we also observe that ICON does not faithfully reconstruct the face or facial features of a clothed human mesh in accordance with the given input RGB image (see first row in the Fig. 6). Instead, in many cases, the facial features of ICON's generated meshes seems to bear resemblance with the facial features of a SMPL-X mesh.

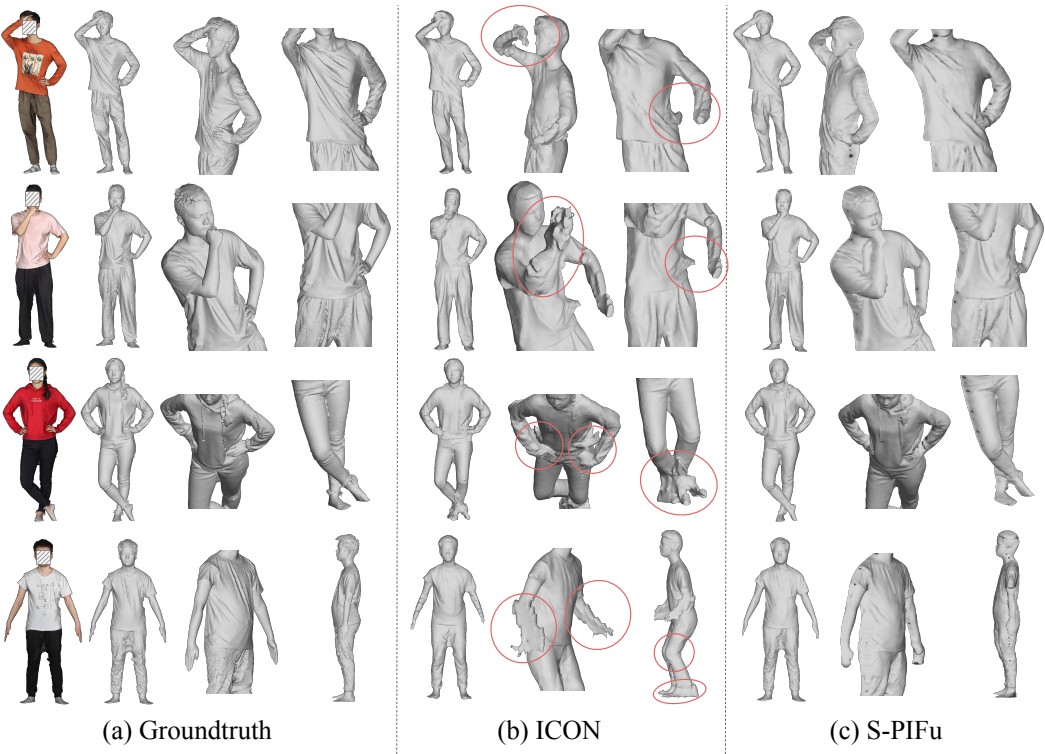

(a) Groundtruth        (b) ICON        (c) S-PIFu

Figure 5: Qualitative evaluation of ICON and S-PIFu. Both models are trained and evaluated on the entire 526 3D scans from THuman2.0 dataset with a 80-20 train-test split.

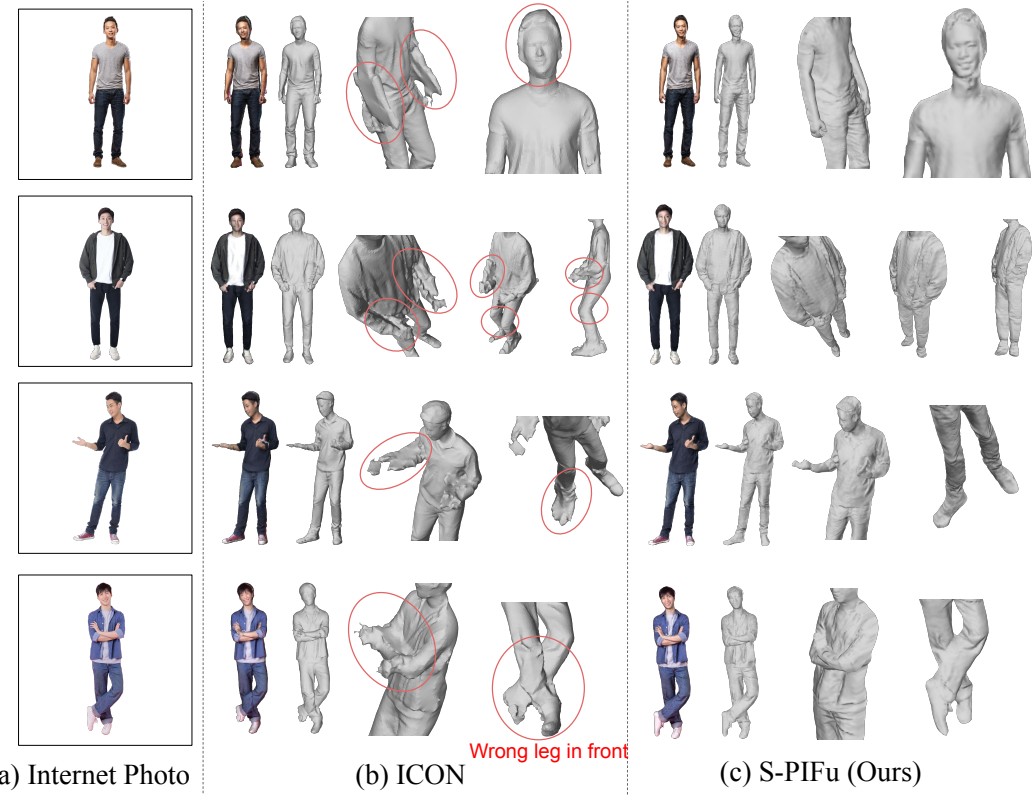

| (a) Internet Photo | (b) ICON | (c) S-PIFu (Ours) |

Figure 6: Comparison of ICON and S-PIFu on real Internet photos sourced from Shutterstock. Both models are trained and evaluated on the entire 526 3D scans from THuman2.0 dataset with a 80-20 train-test split.

Table 2: Quantitative evaluation of S-PIFu if we use Barycentric Interpolation instead of Simple Face Averaging.

|  | THuman2.0 | | BUFF | |
| --- | --- | --- | --- | --- |
| Methods | CD ($10^{-4}$) | P2S ($10^{-4}$) | CD ($10^3$) | P2S ($10^3$) |
| Barycentric | 2.092 | 1.649 | **1.908** | **1.706** |
| Face Average | **2.035** | **1.629** | 1.923 | 1.726 |

## 6 Discussion of applying barycentric interpolation in Ray-based Sampling

In S-PIFu, we take the face average (i.e. simple average of the 3 face vertices) when computing the depth, normal, normal at rest pose, and blendweight of a cell on the feature maps. In Tab. 2, we show the change in results if we had used barycentric interpolation rather than taking a simple face average. As shown in the table, the difference is not significant. This is expected because of two reasons. First, our feature maps are generated at a relatively high resolution of 512x512. We do not expect the simple face average to deviate too much from values obtained from barycentric interpolation. Second, our S-PIFu's architecture is designed such that the generated 2D feature maps, along with the RGB input image and the frontal and rear normal images, are fed in at the very start of S-PIFu pipeline. This gives the S-PIFu a chance to examine and combine all the information contained in the inputs at an early stage and to eliminate any uncorroborated noisy signal from the information early.

## 7 Motivation behind conditioning vertex positions as input features

The initial motivation behind our paper is to create a new way of combining an estimated SMPL-X mesh with a PIFu (or any pixel-aligned implicit model). During the time of our research, the two

Table 3: Quantitative comparison between the different variants of S-PIFu. ( M = Frontal and Rear Normal Maps. C,B,N = Coordinates information, Blendweight-based labels, Normal information respectively. S-PIFu=PIFu+M+C+B+N)

| | THuman2.0 | | BUFF | |
|---|---|---|---|---|
| Methods | CD ($10^{-4}$) | P2S ($10^{-4}$) | CD ($10^3$) | P2S ($10^3$) |
| PIFu | 10.83 | 6.671 | 3.427 | 3.926 |
| PIFu + M | 3.085 | 2.673 | 1.979 | 1.973 |
| PIFu + M + C | **1.999** | **1.626** | 2.022 | 1.840 |
| PIFu + M + B | 2.595 | 2.079 | 2.050 | 1.866 |
| PIFu + M + N | 2.818 | 2.275 | 2.060 | 1.931 |
| PIFu + M + C + B | 2.199 | 1.725 | **1.877** | **1.650** |
| PIFu + M + C + N | 2.275 | 2.056 | 1.929 | 1.862 |
| PIFu + M + B + N | 2.808 | 2.382 | 1.962 | 1.788 |
| S-PIFu | 2.035 | 1.629 | 1.923 | 1.726 |

leading models for this are the PaMIR (9) and ARCH++ (3). Both models showed that the (x,y,z) coordinates of surface points on the SMPL-like mesh can be an useful input for a pixel-aligned implicit model. PaMIR encodes the (x,y,z) coordinate information by voxelizing a SMPL mesh. ARCH++ does the encoding by sampling 3D points from a SMPL-like mesh and using the 3D points to train a PointNet++ (5). The trained PointNet++ then provides encoded coordinate information to the ARCH++ model.

Inspired by PaMIR and ARCH++, we decided to extract the coordinate information of the SMPL-X mesh. The coordinate information, for our case, is computed by taking the simple average of the (x,y,z) values of the three face vertices (together forming a face) on the SMPL-X mesh. We do this for every face on the SMPL-X mesh that is hit by a ray during Ray-based sampling.

# 8 Discussion and Comparison between S-PIFu, S-PIFu - C, S-PIFU - B, and S-PIFu - N

With just coordinate information alone (i.e. PIFu + C + frontal and rear normal maps), the model is unable outperform the competing PIFuHD model (7) in the BUFF dataset. It is important for our models to be able to outperform PIFuHD in the BUFF dataset (and not just the test examples in THuman2.0 dataset) because it shows that our model can generalise to a different data distribution. A different data distribution has more variability in the human scans and may behave very differently from the THuman2.0 dataset. Thus, we believe that a model that has much more potential information (e.g. normal and human parsing feature maps) about an unseen test example from a BUFF dataset has a greater chance of performing well. In Tab. 3, we actually show that combining C with N, B, or both is very useful in improve the model's performance in the BUFF dataset.

Moreover, we believe that our model with coordinate information (C) only, by doing well in THuman2.0 but not in BUFF, may have learned something that is only relevant to the THuman2.0 dataset and not relevant in other datasets.

To add on to the discussion, we show a comparison between S-PIFu, S-PIFu - C (i.e. S-PIFu without C), S-PIFU - B (i.e. S-PIFu without B), and S-PIFu - N (i.e. S-PIFu without N) in Tab. 3. The results show that S-PIFu, while not the best among the variants any of the columns, is the most consistent high-performer for all of the metrics and datasets. None of the variants is able to do well in both the THuman2.0 and BUFF test sets.

S-PIFu, which is given much more information (i.e. C, B, and N) about the training examples, will have a greater chance of learning features that are generalisable to both the THuman2.0 test set and the unseen BUFF dataset, and this is supported by Tab. 3, which shows that when C is combined with B or N, the model's performance on BUFF dataset improves.

# 9    Motivation behind using body parts orientation in S-PIFu

When a body part, such as a human thumb, is being rotated, folded or twisted, its appearance in a 2D RGB photograph or a 3D space will change significantly. A thumb will certainly has a different appearance from an index finger, but that is not what our body parts orientation information is trying to solve. Instead, body parts orientation information helps to give our model a hint of how the thumb is being oriented (is it rotated, folded, or twisted?). For example, if we assume that in the thumb of a SMPL-X mesh is made up of 40 faces, then body parts orientation information will give information on the normal vectors of these 40 faces. If the SMPL-X mesh is giving a thumb-up sign to the camera (where the camera is facing the z-direction), the camera will be seeing the lateral/side-view of a thumb and most of the 40 normal vectors is facing towards or away from the x-direction. However, if the SMPL-X mesh is waving 'Hi' to the camera, then we will be seeing the rear view of the thumb (the side that is opposite the literal thumbnail) and most of the 40 normal vectors is facing towards or away from the z-direction. The 40 normal vectors, which is part of our body parts orientation information, will give a hint of where the thumb is facing. Getting the orientation of the thumb (e.g. where the thumb is facing) is important because it will affect how the thumb will look like and our model, which is designed to reconstruct a human mesh that includes the thumb, will definitely appreciate any information that can help it reconstructs a human mesh correctly.

# 10    Comparison between using Discrete Human Parsing labels and using Blendweight-based Soft Human Parsing labels

This section builds on *Section 3.2* and *Section 4.3.2* in the **main paper**.

In order to compare between using Blendweight-based soft human parsing labels and using discrete human parsing labels, we train two separate PIFu models. The two PIFu models will both be given the inputs of RGB image, frontal normal map, and rear normal map. In addition, the first PIFu takes in Blendweight-based human parsing labels as an additional input. On the other hand, the second PIFu takes in discrete human parsing labels as an additional input.

After both models are trained, we compare them qualitatively in Fig. 7. We find that, compared to discrete human parsing labels, blendweight-based human parsing labels can better tackle the problems of floating artefacts and unnaturally elongated human parts.

Quantitatively, we can compare the two models using Tab. 4. We find that using blendweight-based human parsing ('PIFu + M + B') outperformed using discrete human parsing labels ('PIFu + M + D') in all metrics. This supports our decision to use blendweight-based human parsing labels rather than discrete human parsing labels.

Table 4: Quantitative evaluation of a PIFu trained using discrete human parsing labels (PIFu + M + D) and a PIFu trained using blendweight-based soft human parsing labels (PIFu + M + B) in the THuman2.0 test set and BUFF dataset. M refers to frontal and rear normal maps.

| | THuman2.0 | | | BUFF | | |
|---|---|---|---|---|---|---|
| Methods | CD ($10^{-4}$) | P2S ($10^{-4}$) | Normal | CD ($10^3$) | P2S ($10^3$) | Normal |
| PIFu + M + D | 2.860 | 2.483 | 1.419 | 2.283 | 2.112 | 1.561 |
| PIFu + M + B | **2.595** | **2.079** | **1.411** | **2.050** | **1.866** | **1.559** |

# 11    Comparison between using just current orientation information and using both initial and current orientation information in a PIFu

This section builds on *Section 3.3* and *Section 4.3.3* in the **main paper**.

In *Section 3.3* of the **main paper**, we describe how we extract both the current orientation of body parts and the initial orientation of the body parts from a SMPL-X model. We then incorporate the extracted information into a PIFu. In this section, we will evaluate the necessity of using the initial orientation of the body parts.

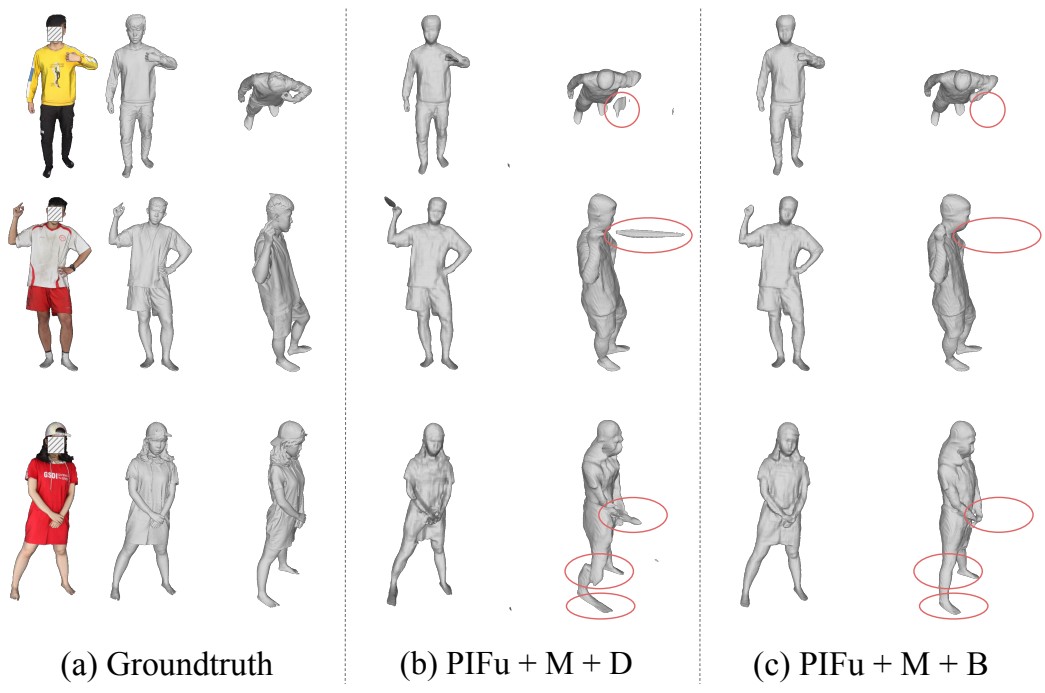

| (a) Groundtruth | (b) PIFu + M + D | (c) PIFu + M + B |

Figure 7: Qualitative evaluation of a PIFu trained using discrete human parsing labels (PIFu + M + D) and a PIFu trained using blendweight-based soft human parsing labels (PIFu + M + B).

We train two separate PIFu models that are each given the inputs of RGB image, frontal normal map, and rear normal map. The first PIFu takes in body part orientation information extracted using Ray-based Sampling as an additional input. The second PIFu takes in the same additional input but without the initial orientation information (only the current orientation information).

After both models are trained, we compare them qualitatively in Fig. 8. We find that when the PIFu is given the initial orientation information, it can reduce the occurrence of floating artefacts and unnaturally elongated human parts.

Quantitatively, we can compare the two models using Tab. 5. We find that incorporating both initial and current body part orientation information ('PIFu + M + N') outperforms incorporating just current body part orientation information ('PIFu + M + N w/o initial orientation') in all but one of the metrics. This supports our decision to incorporate both initial and current body part orientation information into our S-PIFu.

Table 5: Quantitative evaluation of a PIFu trained using Body part orientation information (PIFu + M + N) and a PIFu trained using the same inputs but without the *initial* body part orientation information. M refers to frontal and rear normal maps.

| | THuman2.0 | | | BUFF | | |
|---|---|---|---|---|---|---|
| Methods | CD ($10^{-4}$) | P2S ($10^{-4}$) | Normal | CD ($10^3$) | P2S ($10^3$) | Normal |
| PIFu + M + N | 2.818 | **2.275** | **1.416** | **2.060** | **1.931** | **1.557** |
| PIFu + M + N w/o initial orientation | **2.500** | 2.380 | 1.417 | 2.362 | 2.341 | **1.557** |

## 12 Comparison between a PIFu with coordinate information via Ray-based Sampling and an ARCH++

This section builds on *Section 3.1* and *Section 4.3.1* in the **main paper**.

In this section, we want to compare between using Ray-based Sampling (RBS) and ARCH++ (3) fairly. This means that we have to limit RBS to extract only coordinate information (i.e. no blendweight-based human parsing labels and body part orientation information).

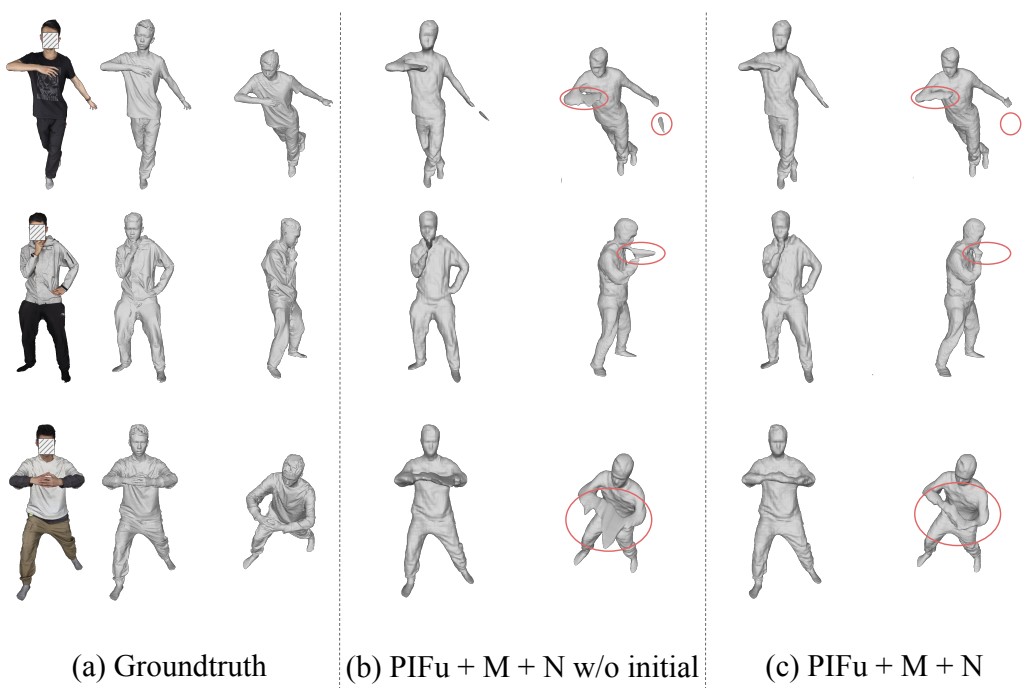

(a) Groundtruth     (b) PIFu + M + N w/o initial     (c) PIFu + M + N

Figure 8: Qualitative evaluation of a PIFu trained using Body part orientation information (PIFu + M + N) and a PIFu trained using the same inputs but without the *initial* body part orientation information.

We train a PIFu model that is given the inputs of RGB image, frontal and rear normal maps, and coordinates information extracted from a SMPL-X model using RBS. Similar to ARCH++, we refine the meshes generated by the PIFu using normal maps. The mesh refinement technique used here is the same as the one used by ARCH++ in (3).

After both models are trained, we compare them qualitatively in Fig. 9. We find that ARCH++ tends to produce missing or unnatural-looking body parts. In contrast, a PIFu trained with coordinates information from RBS ('PIFu + M + C') does not suffer from these problems. As we explained in the **main paper**, we believe that the problems encountered by ARCH++ are caused by its over-reliance on having a near-perfect mapping between the canonical and posed spaces, which is often infeasible to attain due to the varied clothing of human meshes.

Quantitatively, we can compare the two models using Tab. 6. We find that a PIFu that is incorporated with coordinates information extracted using RBS ('PIFu + M + C') is able to significantly outperform ARCH++ in all but one metrics. While ARCH++ has a better P2S score than 'PIFu + M + C' in the BUFF dataset, the score difference is marginal. This supports our argument that RBS is able to utilize coordinates information from a parametric body model more effectively than the ARCH++ model.

Table 6: Quantitative evaluation of a PIFu trained using coordinated information extracted from Ray-based Sampling (PIFu + M + C) and an ARCH++. M refers to frontal and rear normal maps.

| Methods | THuman2.0 | | | BUFF | | |
| --- | --- | --- | --- | --- | --- | --- |
| | CD ($10^{-4}$) | P2S ($10^{-4}$) | Normal | CD ($10^3$) | P2S ($10^3$) | Normal |
| PIFu + M + C | **1.999** | **1.626** | **1.413** | **2.022** | 1.840 | **1.536** |
| ARCH++ | 3.670 | 1.979 | 2.131 | 3.050 | **1.811** | 3.436 |

## 13   More Implementation details and the Source Code

The training of S-PIFu is similar to that of the PIFu model designed by Saito *et al.* (6). We follow Saito's update on the PIFu model in (7) and first train a Normal Predictor to predict a pair of

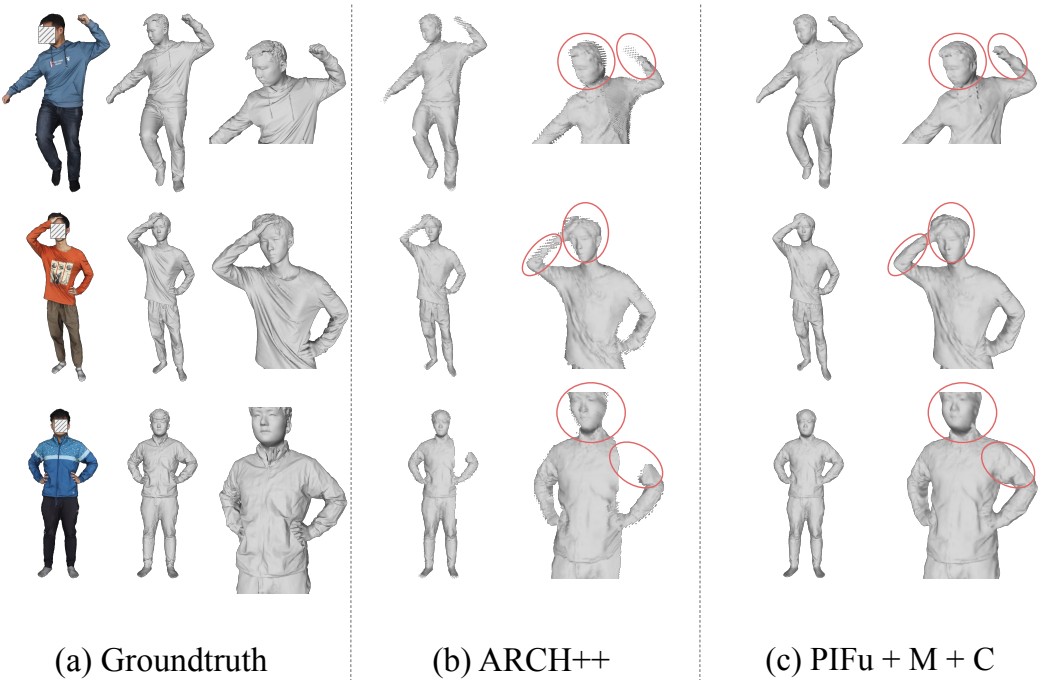

| (a) Groundtruth | (b) ARCH++ | (c) PIFu + M + C |
|---|---|---|

Figure 9: Qualitative evaluation of a PIFu trained using coordinated information extracted from Ray-based Sampling (PIFu + M + C) and an ARCH++ (3).

1024x1024 frontal and rear normal maps from a 1024x1024 RGB image. The architecture of the Normal Predictor is the same as the one used in (7).

S-PIFu is trained with the following inputs: 1. A single RGB image 2. A pair of predicted frontal and rear normal maps 3. A set of 2D feature maps (consists of coordinate information, blendweight-based soft human parsing labels, and body part orientation information). Each of these inputs (1. to 3.) has a resolution of 512x512.

Like (6; 7; 2), we use spatial sampling scheme to train S-PIFu, and we use 8000 sample points for each RGB image.

We use a RMSprop optimizer and a learning rate of $1e-3$. The encoder of the PIFu uses a stacked hourglass network (4) with 4 stacks. The MLP of the PIFu has dimensions of (257, 1024, 512, 256, 128, 1), with skip connections at the third, fourth, fifth layers.

The full implementation details of S-PIFu is contained in our code (https://github.com/kcyt/SPIFu).

As for hardware, we train all the models using a Tesla V100 GPU.