# OpenReview forum: "S-PIFu: Integrating Parametric Human Models with PIFu for Single-view Clothed Human Reconstruction"
_NeurIPS.cc/2022/Conference — NeurIPS 2022 Accept_

### Official Review · Reviewer_FRso · 2022-07-08

**Rating:** 4
**Confidence:** 4
**Soundness:** 3 good
**Presentation:** 3 good
**Contribution:** 3 good

**Summary:**

This paper proposes a new method for single-view clothed human reconstruction by incorporating a parametric body model into a pixel-aligned implicit model. The proposed method has a ray-based sampling method to transform a parametric model into a set of feature maps, proposes blendweight-based soft human parsing labels to improve the structural fidelity, and a method to extract body part orientation information from a parametric model to improve the reconstruction quality.

**Questions:**

Please see the above section.

**Limitations:**

It seems not.

**Strengths And Weaknesses:**

Strengths.

1. A ray-based sampling method is proposed to transform a parametric model into a set of pixel-aligned 2D feature maps.
2. Blendweight-based soft human parsing labels are used.
3. A method that extract and capitalize on body part orientation information from a parametric model is proposed.

Weakness.

The main idea of this paper is to enrich the 2D features with more information derived from SMPL-X parametric body model, before feeding the 2D features into PIFU model for reconstruction. However, three proposed methods seem more like tricks than novel techniques to me.

I am not sure why estimation of body parts orientation can help correct errors caused by shape variations of a body part (explained in L175-176).

The proposed method relies on [16] to generate the SMPL-X parametric body model. What happens if [16] fails?

Based on Table 1, it seems that PIFU+C performs best on the THuman2.0 dataset than the S-PIFu method. Besides, on the BUFF dataset, all three ablated versions (PIFu+C, B, N) perform (slightly) worse than PIFu or PIFuHD. Can the authors explain this?

---

> ### Author Response · Authors · 2022-08-02
> **Comment for Reviewer 3**
>
> Thank you for your helpful comments.
>
> Please refer to Section 1.3 of NIPS_Supplementary_Materials_SPIFu_After_Rebuttal.pdf (i.e. our updated supplementary materials).
>
> We will include all rebuttal materials in either the main paper or the supplementary materials should our submission gets accepted.

---

> > ### Author Response · Authors · 2022-08-06
> > **Update for Reviewer 3**
> >
> > We have added the full results comparing ICON and S-PIFu when both models are trained and evaluated on the entire THuman2.0 dataset. Please refer to the newly added section 1.4 in our supplementary materials.

---

### Official Review · Reviewer_sxwr · 2022-07-09

**Rating:** 6
**Confidence:** 5
**Soundness:** 3 good
**Presentation:** 3 good
**Contribution:** 3 good

**Summary:**

This paper introduces S-PIFu which incorporates a SMPL-X model into PIFu for single-view clothed human reconstruction. Specifically, the authors propose a ray-based sampling technique to transform a SMPL-X model (fitted to the input image) into a set of 2D feature maps embedding coordinate, parsing label, and surface normal information. These feature maps are concatenated with the input image (together with the estimated front and rear normal maps) and fed to the encoder of PIFu. Occupancy prediction then follows the normal pipeline of PIFu. Both quantitative and qualitative evaluations show SOTA results.

**Questions:**

In addition to the points listed as cons above,
- The original PIFu does not take estimated front and rear normal maps as input. They are only being used in PIFuHD. Please clarify your implementation of PIFu in the comparison and update the results if necessary.
- Average coordinates of the face vertices are being used in producing the coordinate feature map. It is actually possible to compute the exact intersection of a ray with a face. How does this compare with using average coordinates?
- Similarly, average normal is computed from the face vertices in producing the normal feature map. How does this compare with simply using normal of a face (triangle)? Alternatively, normal at the point of intersection can be computed using some shading model (e.g., Phong shading). How does this compare with using an average normal?
- Following the last two comments, it seems average parsing label (although not explicit mentioned) is computed from the face vertices in producing the parsing feature map. It is possible to compute an interpolated parsing label by modifying a shading model to work with parsing labels (e.g., Phong shading). How does this compare with using an average paring label?
- Table 1 shows that S-PIFu performs worse than PIFu+C on THuman2.0. This suggests that most improvement is actually brought by the coordinate information, and the parsing and normal information may actually harm the occupancy prediction (due to the increased channel number?). Further analysis and discussion are needed. It might be desirable to show the comparison between S-PIFu, S-PIFu - C, S-PIFU - B, and S-PIFu - N in the ablation study. Since the estimated front and rear normal maps are not used in PIFu, another variant to be considered is S-PIFu - 'front & rear normal'.


**Limitations:**

No specific limitation and negative societal impact need to be addressed.

**Strengths And Weaknesses:**

Pros:
+ The proposed ray-based sampling technique for transforming a SMPL-X model into a set of 2D feature maps sounds novel. It provides a simple and effective way to encode both geometric and semantic information of the 3D parametric model. This 2D feature map representation works seamlessly with the PIFu framework.
+ The proposed soft parsing labels based on blending weights sounds novel. It is more informative than discrete parsing label and can help to improve the labels around the boundary between two body parts.
+ Other than spatial information (i.e., 3D coordinates) which have been exploited by previous works, this work also extract human parsing labels and surface normals from the parametric model which provide further semantic and geometric information for occupancy prediction.
+ Quantitative and qualitative evaluations show the proposed method outperforms PIFu, PIFuHD, and ARCH++.
+ Experiments are included to demonstrate the improvement brought by the coordinate, parsing label, and surface normal feature maps, respectively, over PIFu.
+ The paper is well written and easy to follow.

Cons:
- The authors argue that ARCH++ depends on having a near-perfect mapping between the canonical and posed spaces. Note that such a mapping is defined by the skinning weights and hence it depends on the accuracy of the model fitting. Likewise, this work constructs 2D feature maps from the fitted SMPL-X model and therefore it also depends on the accuracy of the model fitting. However, there is no analysis showing how the model fitting accuracy/error will affect the reconstruction.
- Unlike ARCH++ where pixel-aligned features (256 channels) and spatial features (96 channels) are computed independently before concatenation for occupancy prediction, 2D feature maps (~198 channels) are constructed from the fitted SMPL-X model which are concatenated with the input image (3 channels) and the estimated front and back normal maps (6 channels), and fed to the encoder of PIFu. Note that only 3 out of ~207 channels in the input come from the RGB image. This suggests the occupancy prediction may heavily depend on the fitted SMPL-X model. There is no analysis on any possible bias on the source of input.
- Meshes with extreme self-occlusion are excluded in training. This reduces the number of usable meshes in THuman2.0 from 526 to 362. With reduced training data, the performance of the models (both the proposed model and existing models for comparison) will be affected (to different extends), particularly in handling self-occlusion.
- For each mesh in THuman2.0, only 10 RGB images at different yaw angles have been used in training. Previous works typically use 360 images around each models for training. This difference in training setting makes it not possible to directly/easily compare the quality of models with previous works. For instance, the results of ARCH++ presented in this paper look much worse than in the original paper.

---

> ### Author Response · Authors · 2022-08-02
> **Comment for Reviewer 2**
>
> Thank you for your kind comments.
>
> Please refer to Section 1.2 of NIPS_Supplementary_Materials_SPIFu_After_Rebuttal.pdf (i.e. our updated supplementary materials).
>
> We will include all rebuttal materials in either the main paper or the supplementary materials should our submission gets accepted.

---

> > ### Author Response · Authors · 2022-08-06
> > **Update for Reviewer 2**
> >
> > We have added the full results comparing ICON and S-PIFu when both models are trained and evaluated on the entire THuman2.0 dataset. Please refer to the newly added section 1.4 in our supplementary materials.

---

### Official Review · Reviewer_49L7 · 2022-07-11

**Rating:** 3
**Confidence:** 5
**Soundness:** 2 fair
**Presentation:** 1 poor
**Contribution:** 2 fair

**Summary:**

This paper presents several input features derived from parametric human model to PIFu-like 3D reconstruction task of clothed humans. At each pixel from the input view, this work propose to compute the multiple ray intersection with the parametric body model and extract positions, skinning weights, and surface normal at the intersections as auxiliary pixel-aligned features. The experiments show that the proposed features significantly improve the robustness and accuracy of reconstruction over prior methods in THuman2.0 and BUFF datasets.

**Questions:**

- Please address the concerns above.
- After reading through the paper, I still couldn’t figure out what “S” in S-PIFu stands for. Please clarify.
- Since intersection locations can be obtained via barycentric interpolation, why is simply averaging of 3 vertices used? Does the performance improve by taking barycentric interpolation?
- What value is inserted if the ray does not hit surface?
- What is the motivation behind conditioning vertex positions as input features? Please elaborate.

Other comments:
- Citation format is not the official one.
- L5: blendweight-based soft human parsing label is merely skinning weights . Please use common term.
- L87: The method that proposes to use front and back normals are PIFuHD not PIFu.
- L168: principal component analysis

**Limitations:**

Neither limitation nor its societal impact is discussed in the paper.

**Strengths And Weaknesses:**

The strengths of the paper can be summarized as follows:
- The simplicity of the proposed method is a great advantage. Especially the plug and play nature of the proposed feature encoding allows us to boost the performance regardless of the baseline method (e.g, PIFu, PIFuHD).
- The experimental results support the effectiveness of the proposed approach.

On the other hand, the paper has the following weaknesses:
- It is not clear how this approach performs for the pixels that do not belong to SMPLX body. Non-overlapped regions are quite common for clothed human as silhouette of clothed humans is larger than minimally clothed SMPLX. Intuitively, the proposed approach provide no information for these pixels.
- Related to this, the proposed approach seems to require very accurate underlying SMPLX fitting. However, as PaMIR and ICON mentioned, off-the-shelf body pose estimation networks do not provide very accurate fitting and thus additional optimization is required. It is not clear how well the proposed approach handles inaccurate pose parameters. Qualitative results from images in the wild and evaluation with noise perturbed pose parameters are highly recommended.
- The paper misses one important work, ICON, which improves robustness over PIFu by conditioning SMPL body. Please discuss and compare.

ICON: Implicit Clothed Humans Obtained From Normals
Yuliang Xiu, Jinlong Yang, Dimitrios Tzionas, Michael J. Black; Proceedings of the IEEE/CVF Conference on Computer Vision and Pattern Recognition (CVPR), 2022, pp. 13296-13306

Despite its promising performance, I think the paper is not ready for publication due to its incomplete evaluations and comparisons. Also the illustration and exposition can be further improved.

---

> ### Author Response · Authors · 2022-08-02
> **Comment for Reviewer 1**
>
> Thank you for your strong comments.
>
> Please refer to Section 1.1 of NIPS_Supplementary_Materials_SPIFu_After_Rebuttal.pdf (i.e. our updated supplementary materials).
>
> We will include all rebuttal materials in either the main paper or the supplementary materials should our submission gets accepted.

---

> > ### Author Response · Authors · 2022-08-06
> > **Update for Reviewer 1**
> >
> > We have added the full results comparing ICON and S-PIFu when both models are trained and evaluated on the entire THuman2.0 dataset. Please refer to the newly added section 1.4 in our supplementary materials.

---

> > > ### Comment · Reviewer_49L7 · 2022-08-08
> > > **Re: Comment for Reviewer 1**
> > >
> > > Thanks for the detailed responses as well as additional experiments to address the concerns!
> > >
> > > After reading other reviews and rebuttal, while my initial concerns are well addressed, I am now more concerned about negligible improvement by adding blendweights and surface normal as R3 mentioned. In fact, Tab. 8 in Supp. Mat shows that PIFu + C already performs nearly state-of-the-art performance without B or N, and it's not clear at all if B and N offer any conclusive contributions beyond adding C. As the performance itself is impressive, I would recommend making the paper simpler by focusing on the ray-based sampling as a key contribution and removing specific feature engineering from the main contributions unless adding each feature makes significant improvement. For this reason, I keep my initial recommendation of reject while encouraging to resubmit to a future conference/journal.

---

> > > > ### Author Response · Authors · 2022-08-09
> > > > **Re: Re: Comment for Reviewer 1**
> > > >
> > > > Thank you, we take your comments very seriously. But ray-based sampling alone (i.e. PIFu + M + C) is not sufficient for a paper because it is only able to cleanly outperform the SOTA (PIFuHD) in the THuman2.0 dataset and not in the BUFF dataset (See below; lower values are better for all metrics shown).
> > > >
> > > > |                     |\|| THuman2.0  || \| |  BUFF      ||
> > > > |:-------------:|:-------------:|:-------------:|:-------------:|:-------------:|:-------------:|:-----:|
> > > > |  **Method**    |\|    |   **CD**  (${10^{-4}}$)   | **P2S** (${10^{-4}}$) | \|| **CD** (${10^{3}}$) | **P2S** (${10^{3}}$) |
> > > > |PIFuHD  |\| | 2.758 | 2.215  |  \|| 1.964 | 1.845 |
> > > > |PIFu + M + C  |\| | 1.999 | 1.626 |\| | 2.022 | 1.840 |
> > > > |S-PIFu (i.e. PIFu + M + C + B + N)  |\| | 2.035 |  1.629 | \|| 1.923 | 1.726 |
> > > >
> > > > Given that the models are all trained on the THuman2.0 train set, it is very important that the our models do well in a dataset with an entirely new data distribution (i.e. BUFF dataset) as it tests their ability to generalize correctly.
> > > > The table above shows that adding B and N is pivotal in ensuring that our final model (S-PIFu) is able to significantly outperform the SOTA (PIFuHD) in both datasets, without which we cannot really make a case to publish a paper.
> > > >
> > > > Quantitatively, if we choose to exclude B and N from S-PIFu, then the CD and P2S in BUFF dataset would worsen by 5.1% and 6.6% respectively. The slight tradeoff is that the CD and P2S in THuman2.0 test set would improve by 1.7% and 0.18% respectively.

---

### Author Response · Authors · 2022-08-08
**Soliciting Feedbacks**

Dear Reviewers,

It is the last day of the Author-Reviewer discussion period. We would like to ask the reviewers if any part of our rebuttal requires more attention or if the reviewers have any additional concern regarding the paper?

Best regards, \
Authors

---

### Meta-Review · Area_Chair_x5jY · 2022-08-27

**Recommendation:** Accept
**Confidence:** Certain

**Metareview:**

All reviewers consider this a novel and effective contribution to the increasingly important subfield of 3D human reconstruction, particularly from unusual poses, or, as exposed in the rebuttal, with loose clothing.

The key technical questions of the reviewers (both positive and negative) were about dependence on accurate pose parameters, and dependence on accurate surface fits, which would be incorrect for e.g. baggy clothing.  The rebuttal does a thorough and convincing job in exploring these questions,

Reviewer FRso says:
 - The three proposed methods "seem more like tricks".  This does not refute their novelty - that would be achieved by pointing to specific prior art.
 - "What happens if [16] fails?"  This is now well answered in the rebuttal, and the answer is satisfactory
 - "ablated versions perform worse" - the new tables show this can be the case on some datasets, but not on others.  Of course, it would be ideal if some mechanism could downweight these contributions where appropriate, but that is not a task for this paper.

Reviewer 49L7 says:
 - " It is not clear how this approach performs for the pixels that do not belong to SMPLX body".  Now answered well in the rebuttal.
 - "seems to require very accurate underlying SMPLX fitting".  Now answered well in the rebuttal.
 - "misses one important work, ICON".  As the rebuttal notes, the code for this work was released very shortly before the deadline.  The rebuttal is careful to give the timeline for the code release, rather than just using the CVPR conference date.
The rebuttal also includes a preliminary but useful comparison to ICON, showing that in fact the paper outperforms ICON (when trained on similar data), but also noting that they are very different architectures, which further argues for both being exposed to the community.
I agree with the authors that the later objections of R1 are "moving the goalposts".  I would not necessarily dismiss those later objections if they were fundamental, but again, the rebuttal answers them convincingly.

Reviewer sxwr was overall in favour of accept, but had some queries, again well responded to in the rebuttal.


**Award:**

No

---

### Decision · Program_Chairs · 2022-09-14

Accept